# Structural insight into the bulge-containing *KRAS* oncogene promoter G-quadruplex bound to berberine and coptisine

Kai-Bo Wang [1,3] ✉, Yushuang Liu[1,3], Jinzhu Li[1], Chengmei Xiao[1], Yingying Wang[1], Wei Gu[2], Yipu Li[1], Yuan-Zheng Xia[1], Tingdong Yan[2], Ming-Hua Yang[1] & Ling-Yi Kong [1] ✉

KRAS is one of the most highly mutated oncoproteins, which is overexpressed in various human cancers and implicated in poor survival. The G-quadruplex formed in *KRAS* oncogene promoter (*KRAS*-G4) is a transcriptional modulator and amenable to small molecule targeting. However, no available *KRAS*-G4-ligand complex structure has yet been determined, which seriously hinders the structure-based rational design of *KRAS*-G4 targeting drugs. In this study, we report the NMR solution structures of a bulge-containing *KRAS*-G4 bound to berberine and coptisine, respectively. The determined complex structure shows a 2:1 binding stoichiometry with each compound recruiting the adjacent flacking adenine residue to form a "quasi-triad plane" that stacks over the two external G-tetrads. The binding involves both $\pi$-stacking and electrostatic interactions. Moreover, berberine and coptisine significantly lowered the *KRAS* mRNA levels in cancer cells. Our study thus provides molecular details of ligand interactions with *KRAS*-G4 and is beneficial for the design of specific *KRAS*-G4-interactive drugs.

*KRAS* is justifiably renowned as one of the most highly mutated oncogenes in human cancers and implicated in poor survival, particularly in pancreatic (~95%), colorectal (~41%), and lung (~32%) cancers[1,2]. The KRAS protein functions as a small switch signaling GTPase that toggles between an inactive, guanosine-diphosphate (GDP)-bound state and an active, guanosine-triphosphate (GTP)-bound form[1,3]. Mutations of KRAS impair its ability to hydrolyze GTP and result in abnormally high levels of GTP-bound KRAS, which lead to the continuous activation of *KRAS* signaling pathways and uncontrolled cell proliferation, migration, transformation, and survival[1,4]. Although KRAS is a highly pursued therapeutic target, direct inhibition of mutant KRAS oncoprotein has proven to be challenging in precision oncology for nearly 40 years[1]. The high intracellular GTP/GDP concentration and their picomolar binding affinity to the KRAS, coupled with the lack of small-molecule binding pockets on the surface of KRAS oncoprotein, lead to a long-standing notion that mutant KRAS is "undruggable"[1,4–6]. Encouragingly, recent advances in medicinal chemistry have discovered an allele-specific KRAS(G12C) covalent inhibitor, sotorasib (AMG-510), that showed clinical benefit for cancer patients and was conditionally approved by the FDA in 2021[1,7–9]. However, the KRAS(G12C) only covers a small proportion of KRAS mutations, which were found in ~13% of lung adenocarcinomas, 3% of colorectal cancer, and at a lower frequency in other cancers (less than 1% in pancreatic cancer)[2,4,6]. Additionally, the diverse and high-frequency acquired resistances further limit the efficacy of KRAS(G12C) treatment[4,10]. Furthermore, the amplification of wild-type KRAS has been a secondary means for KRAS activation across a number of human cancers[11,12]. Therefore, it is essential to develop new types of *KRAS* signaling inhibitors.

[1]Jiangsu Key Laboratory of Bioactive Natural Product Research and State Key Laboratory of Natural Medicines, Department of Natural Medicinal Chemistry, China Pharmaceutical University, Nanjing 210009, China. [2]School of Life Sciences, Shanghai University, Shanghai 200444, China. [3]These authors contributed equally: Kai-Bo Wang, Yushuang Liu. ✉e-mail: kbwang@cpu.edu.cn; cpu_lykong@126.com

G-quadruplexes (G4s) are unique DNA secondary structures that consist of the stacking of Hoogsteen hydrogen-bonded multiple G-tetrads, which are stabilized by the coordination of $K^+$ or $Na^+$[13–16]. The human genomic DNA has lots of G4-forming motifs while endogenous G4s formation is substantially associated with highly transcribed genes[14,17–20]. DNA G4 structures are extremely concentrated in the key regulation region of gene promoters implicated in cancer and neurodegenerative disorders[13,17,21]. The dynamic formation of DNA G4s in live human cells is cell-cycle-dependent, which can be trapped in the folded states by G4-specific small molecules for gene expression inhibition[22–26]. G4 structures have thus emerged as distinct DNA targets for cancer therapy, especially as alternative strategies for those "undruggable" proteins, such as MYC and KRAS[13,27–31].

The promoter region of the human *KRAS* oncogene has a critical GC-rich nuclease hypersensitive element (NHE), which can form a DNA G4 structure (*KRAS*-G4)[28]. *KRAS*-G4 has shown to be a transcriptional modulator by interplaying with several nuclear proteins, which is amenable to small-molecule targeting[32–34]. Small molecules can compete with nuclear proteins to bind the *KRAS*-G4 for *KRAS* gene expression inhibition[35]. To date, a number of *KRAS*-G4 interactive small molecules (ligands) have been reported, together with several types of free *KRAS*-G4 structures[36–43]. However, no available *KRAS*-G4–ligand complex structure has yet been determined, which seriously hinders the structure-based rational design of *KRAS*-G4 targeting drugs. Therefore, much effort is needed to find suitable *KRAS*-G4 binding ligands and to determine their complex structures.

Natural products have long been a trove of structurally diverse compounds and an important source of anticancer drugs[44,45]. Notably, we have recently found that several antitumor alkaloids could effectively bind to DNA G4s, presenting a new mechanism of action for anticancer drug development[46–49]. Among them, berberine and its derivatives have attracted our particular attention as they have diverse pharmaceutical effects, including anticancer, anti-inflammatory, and antimicrobial activities, and present a unique natural skeleton to interact with DNA G4s [47,49,50].

In the current study, we firstly determined the NMR solution structure of a bulge-containing *KRAS*-G4 using a Pu24m1 DNA sequence, which has the critical 2-3 wild-type residues on both the 5′- and 3′-end flankings, the key elements for specific ligand recognition and selectivity. With the obtained *KRAS*-G4 model system, we found that natural alkaloids, berberine (BER) and coptisine (COP), could strongly bind and stabilize the *KRAS*-G4 by using various NMR, CD, and fluorescence experiments. We then determined the atom-level NMR solution structures of *KRAS*-G4 in complex with BER and COP, respectively. The resolved complex structures elucidate the detailed features for the specific recognition of the *KRAS*-G4 by ligands and provide a model system for further developing more potent *KRAS*-G4 targeting drugs. Furthermore, berberine and coptisine significantly stalled the *Taq* DNA polymerase synthesis of the complementary strand DNAs and lowered the *KRAS* mRNA levels in cancer cells.

## Results

### Berberine and coptisine strongly bind and stabilize the *KRAS* oncogene promoter G4

The NHE of the *KRAS* oncogene promoter sequence, namely Pu34 (Fig. 1a), has multiple G-rich tracts, which can form two distinct G4s by using different G-tracts, namely KRAS1234 and KRAS1235 G4s (Fig. 1a and Supplementary Fig. 1)[28,32,39]. Both of these two types of G4s have been demonstrated to be implicated in nuclear protein interactions for *KRAS* oncogene regulation and enable small molecule targeting[32,39,41]. We focused on the KRAS1234, as G4 with shorter loop lengths is speculated to be thermodynamically and kinetically preferred[51,52]. Although the solution structure of KRAS1234 G4 (22R, Fig. 1a) has been determined, the critical element for specific ligand recognition is missing in the 5′-end site, as only an adenine residue is available for the

binding pocket formation[41]. The flanking residues have been well documented to be critical for G4 formation and specific ligand recognition[53,54]. We thus used the Pu24m1 DNA sequence that contains the wild-type TGA residues in the 5′-flanking to better mimic the wild-type *KRAS* promoter G4 (for convenience, the *KRAS*-G4 refers to Pu24m1 DNA afterward).

We firstly examined the binding interactions of berberine and coptisine (Fig. 1b) to the Pu24m1 DNA sequence using the $^1$H-NMR titration experiments in a $K^+$-containing solution. The free Pu24m1 shows 12 imino proton peaks corresponding to three stacked G-tetrads (Fig. 1c). Upon individual addition of berberine and coptisine, almost all imino proton resonances of the free *KRAS*-G4 are upfield shifted, revealing binding of one compound at each outer G-tetrad through end-stacking interactions (Fig. 1c). The binding appears to be in an intermediate-to-fast exchange rate on the NMR time scale because the broadening imino proton peaks are shown at a lower drug ratio of 1:1 and the sharpening at higher drug ratio of 2:1 and 3:1. Notably, a new set of 12 distinct protons are emerged after berberine and coptisine addition, respectively, suggesting the formation of one dominant conformation of the ligand–*KRAS*–G4 complexes. Moreover, a small proportion of DMSO in samples does not affect the NMR spectra quality, as shown in Fig. 1c and Supplementary Figs. 2 and 3.

The binding activity of the two isoquinoline alkaloids to the *KRAS*-G4 was further investigated by CD, EMSA, and fluorescence experiments. The Pu24m1 sequence adopts a parallel G4 topology that is the same as its shorter 22R sequence, as shown by its characteristic CD spectra with a positive band at ca. 264 nm and a negative band at around 242 nm (Figs. 1a and 2a)[55]. Upon the respective addition of two isoquinoline alkaloids, the parallel *KRAS*-G4 topology was maintained with enhanced CD bands at 264 nm (Fig. 2a). BER and COP increased the thermal stability of *KRAS*-G4 by 15 and 19 °C at 2 equivalents, respectively (Fig. 2). Meanwhile, a much smaller $T_m$ change was shown from 2 to 4 equivalents of drug addition, suggesting two dominant binding sites of *KRAS*-G4 for BER and COP. The 2:1 binding stoichiometry was further corroborated in the NMR titration experiment because only minor changes were observed for the imino protons from 2 to 3 ligand equivalents, where the G15-H1 (Fig. 1c) is clearly seen at 3 ligand equivalents. Notably, a ligand:DNA ratio higher than its binding stoichiometry is often needed for ligand binding at the intermediate-to-fast exchange on the NMR timescale[46,50,56]. The higher ligand ratio can push the binding equilibrium toward the formation of a well-resolved complex which shows 12 sharp imino proton peaks. The binding affinity of two isoquinoline alkaloids to the *KRAS*−G4 was determined by a fluorescence-based binding assay. Preformed free *KRAS*-G4 DNA was gradually titrated to a drug solution with a fixed concentration and the induced fluorescence signal was recorded to obtain the dissociation constant ($K_d$) value (Fig. 2b). The determined $K_d$ values are 0.55 μM and 0.50 μM for berberine and coptisine, respectively (Fig. 2c). Moreover, the *KRAS*−G4 in complex with berberine and coptisine was monomeric, as shown by the native EMSA gel (Supplementary Fig. 4). Overall, the above data showed that berberine and coptisine strongly bind and stabilize the *KRAS*-G4 by the formation of well-defined complex structures.

To gain insight into the structural dependencies of berberine and coptisine binding, CD and NMR titration experiments were collected of these two compounds to *MYC*-G4 (parallel)[46], *VEGF*-G4 (parallel)[57], and human telomeric G4s (*Tel*-hybrid1 and *Tel*-hybrid2 G4s)[31,49]. The structural selectivity profiles were evaluated by the $\Delta T_m$ values[58] obtained from the CD melting experiments. The data showed that the two compounds have preferences for the parallel G4s with a higher stabilization effect compared to the telomeric hybrid G4s (Supplementary Figs. 5 and 6). The results were consistent with NMR titration data because berberine and coptisine showed more specific binding to *KRAS*-G4, *MYC*-G4, and *VEGF*-G4, where a well-defined complex was

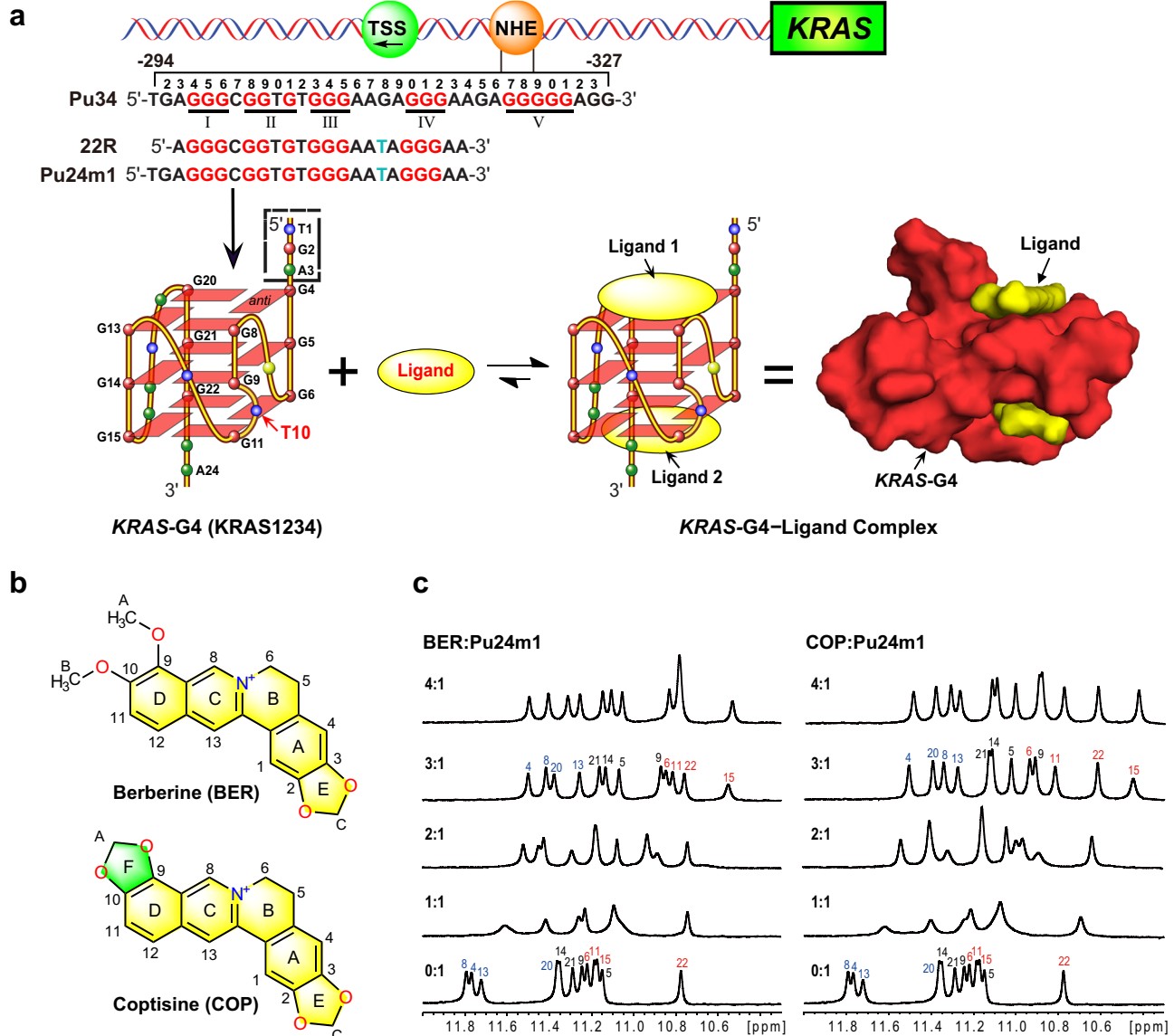

**Fig. 1 | KRAS-G4 and its interaction with berberine and coptisine. a** Schematic of the human *KRAS* gene promoter and the formation of a *KRAS*-G4 as well as its complex with small molecules (ligands). The G4-forming region of the NHE sequence is shown. The G-tracts implicated in *KRAS*-G4 formation are marked in red and mutations in cyan, respectively. **b** Chemical structures of berberine and coptisine with numbering. **c** 1D $^1$H NMR titration of Pu24m1 DNA with berberine and coptisine, respectively, with complete imino proton assignment. Conditions: 150 μM DNA, pH 7, 50 mM K$^+$ solution, 25 °C, DMSO-$d_6$ < 1.5%.

formed at the ligand ratio of 2 or 3, with 12 new emerging imino proton peaks (Fig. 1 and Supplementary Figs. 7 and 8). In contrast, the compounds did not show specific binding to telomeric G4s as the ligand–G4 complex structures did not show 12 well-defined imino proton peaks (Supplementary Figs. 9 and 10).

### NMR solution structure determination of the wild-type bulge-containing *KRAS*-G4

We first determined the high-resolution NMR structure of the free *KRAS*-G4 in the K$^+$ solution, which has a unique T-bulge. The Pu24m1 DNA sequence forms a stable three G-tetrad stacked G4 as shown by 12 well-defined imino proton resonances in the 1D $^1$H-NMR spectrum that is suitable for structure determination (Fig. 1c and Supplementary Fig. 14). A set of 2D NMR spectra, including NOESY, DQF-COSY, and HSQC experiments, at different temperatures and mixing times were collected (Fig. 3 and Supplementary Figs. 15 and 16). All imino, aromatic, and sugar resonances of *KRAS*-G4 were assigned using standard strategies[59] and referring to the reported 22R DNA assignment as both

sequences have the same 3′-end flanking residues (Supplementary Table 2)[41]. The three G-tetrad planes are determined as G4-G8-G13-G20, G5-G9-G14-G21, and G6-G11-G15-G22, which are the same as the 22R G4 with a unique T-bulge (Figs. 1a and 3). All DNA residues adopt anti-glycosidic torsion angles as shown by their medium intensities of H1′-H6/H8 NOE cross-peaks and the corresponding downfield C6/C8 chemical shifts (Fig. 3 and Supplementary Table 2).

The high-resolution NMR solution structure of the *KRAS*-G4 was thus determined using restrained molecular dynamics (MD) simulations based on the distance information extracted from the NOESY experiments (Table 1 and Supplementary Tables 2–5). A total of 745 NOE-derived distances, 48 H-bond, and 24 torsion-angle restraints were used for the simulations. The obtained final ten lowest energy structures were well-converged with a heavy atom root-mean-square deviation (RMSD) of 0.58 ± 0.21 and 0.71 ± 0.21 Å for the G-tetrad core and all residues, respectively (Fig. 4a and Table 1).

The determined parallel *KRAS*-G4 in the K$^+$ solution shows several distinct features compared to the reported 22-RT G4[41], especially for

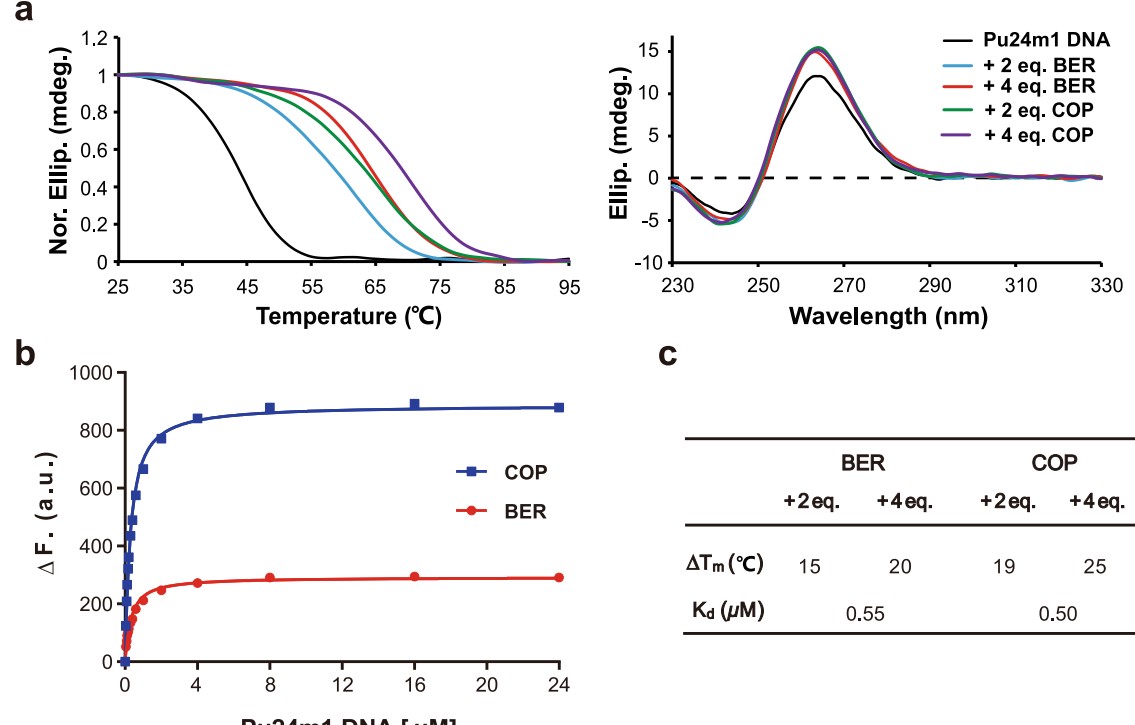

**Fig. 2 | Biophysical characterization of berberine and coptisine binding to the KRAS-G4. a** CD thermal melting curves and CD spectra of Pu24m1 DNA with berberine and coptisine, respectively. Conditions: 20 μM DNA, pH 7, 15 mM K⁺ solution. **b** Fluorescence intensity change of berberine and coptisine upon titration with Pu24m1 DNA, respectively. Conditions: 0.2 μM compound, pH 7, 50 mM K⁺ solution. **c** The determined $\Delta T_m$ and $K_d$ values of berberine and coptisine to the Pu24m1 DNA. The melting temperature ($T_m$) was obtained at the intersection between the median of the fitted baselines and the melting curve. The dissociation constant ($K_d$) was calculated by data fitting with a 2:1 binding equation. The experiments were run in duplicate.

the capping and the 4-nt loop structures. In the 5′-end site, the extended three flanking residues form a well-defined capping structure with G2 and A3 stacking on the 5′-end outer G-tetrad, and their Watson–Crick edges point towards opposite grooves (Fig. 5a and Supplementary Fig. 11a). This arrangement is quite similar to the recently determined MYC-G4 with the same TGA flanking residues[29]. In the 3′-end site, the bulge residue T10 is recruited by the flanking residue A20 and forms a Watson–Crick base pair that stacks on the 3′-tetrad plane, which is supported by NOE cross-peaks of A23H8 to G22H8, of A23H2 to G22H1, G6H1, G6H8, and T10Me, and of T10H1′ to G6H1 (Figs. 3, 5a, Supplementary Figs. 12a, and 16, and Supplementary Table 4). Nevertheless, the A·T base pair is not observed in the 22-RT G4 although the same DNA residues are involved[41]. Moreover, the 4-nt propeller loop is mostly disordered in the 22-RT G4, whereas it now forms a well-defined structured conformation by constructing an ingenious A16-A19-T18 base stacking organization, which is connected by an A17 linker (Figs. 4a, 5a, and Supplementary Fig. 13a). This unique loop structure is well supported by key NOE contacts of A19H8 to A16H8, A17H8, and T18H6, of A16H8 to A17H8, and of T18Me to A17H8 (Supplementary Fig. 16 and Supplementary Table 5). Overall, the DNA backbone of the 4-nt loop is turned outside and electrostatic repulsions between the negatively charged loop phosphates and the G-tetrad core are minimized by the loop's central position (Figs. 4a and 5a). Further, the solvent exposure of the loop residues is reduced by extensive base stacking interaction.

## NMR solution structure determination of the 2:1 berberine–KRAS–G4 complex

The high quality of the 1D ¹H-NMR titration data indicated that a well-defined berberine-KRAS-G4 complex is formed, which is feasible for structure determination (Fig. 1c and Supplementary Fig. 14). Therefore, various 2D NMR spectra of the berberine-KRAS-G4 complex were collected, including HSQC, NOESY, and DQF-COSY (Supplementary Figs. 17–21). Similar NOE cross-peak fingerprints were observed between the free KRAS-G4 and its complex with berberine (Fig. 3, Supplementary Figs. 16, 20, and 21). Complete proton resonances were thus assigned with the aid of the free KRAS-G4 spectra assignment (Supplementary Table 6). Protons of free and bound berberine were determined using 1D and 2D NMR spectra (Supplementary Figs. 19, 27, 29 and Supplementary Table 16). The parallel topology and same G-tetrad organization were maintained in the berberine-KRAS-G4 complex compared to the free KRAS-G4 (Figs. 1a and 4). The largest chemical shift difference ($\Delta\delta$) values were obtained for the H1 protons of the 5′- and 3′-end G-tetrad guanines (Supplementary Fig. 31). Moreover, much larger $\Delta\delta$ values are shown for the flanking residues than the three-loop residues (Supplementary Fig. 32), suggesting that berberine stacks on the two outer G-tetrads.

Numerous inter- and intramolecular NOEs were obtained, which clearly defined the berberine binding sites and the overall complex structures (Fig. 6, Supplementary Figs. 18–21, Table 1, and Supplementary Tables 7–10). The atom-level NMR solution structure of the 2:1 berberine–KRAS–G4 binary complex was then determined using MD simulations. A total of 728 NOE-derived distance restraints were used for the structural calculation (Table 1 and Supplementary Tables 7–10). The binding sites of the berberine at both outer G-tetrads are well-defined and supported by 37 berberine-DNA intermolecular NOE restraints (Figs. 4b, 5b, and 6, Supplementary Table 7). The final ten lowest energy structures are well-converged and show an RMSD of 0.75 ± 0.16 Å for all residues (Fig. 4b, Table 1).

The NMR solution structure shows that the berberine binding induces significant rearrangement for both the 5′-end and 3′-end capping structures (Fig. 5, Supplementary Figs. 11 and 12). At 5′-end, berberine recruits A3 to form a "quasi-triad plan" that stacks over the 5′-end G-tetrad. Compared to the free KRAS-G4, A3 is flipped and the

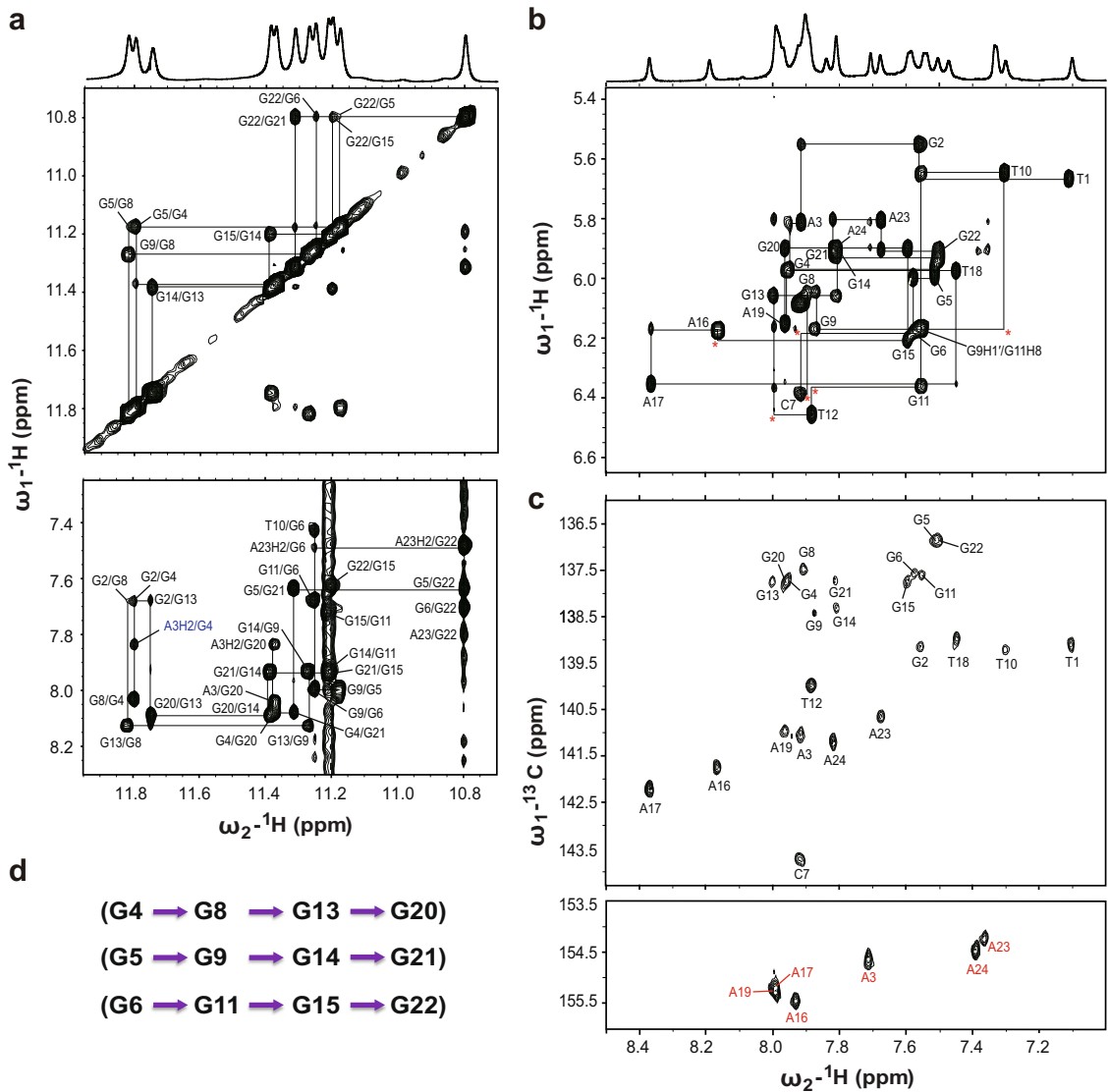

**Fig. 3 | 2D NMR spectra of the free *KRAS*-G4. a** The H1–H1 and H1–H8 regions, and **b** H1′-H6/H8 region from the 2D-NOESY spectra of Pu24m1 DNA in H$_2$O with sequential assignment pathway. Missing connectivity is marked with red asterisks. The cross-peak assigned to the minor species is labeled in blue. **c** H6 − C6/H8−C8

cross-peaks of all bases (black label) and the adenine H2-C2 contacts (red label) with assignments for the Pu24m1 DNA by HSQC experiments. **d** The assigned three G-tetrad planes of *KRAS*-G4 by NMR experiments. Conditions: 1.5 mM DNA, pH 7, 50 mM K$^+$ solution, 25 °C, DMSO-$d_6$ < 3.5%.

Watson-Crick edge now points toward the G-tetrad center, which enables the formation of an A3-berberine plane (Fig. 5a, b). Moreover, with the occupancy of berberine, the G2 is away from the G-tetrad plane and stacks upon the "quasi-triad plane". At 3′-end, berberine binding disrupted the original A·T base pair structure. Instead, an A23-berberine plane is formed by rotating the A23 base for ca. 90°, which maximized the stacking interaction with the 3′-end G-tetrad. Interestingly, the dissociated bulge base T10 partially covers berberine and contributes to the formation of a 3′-end binding pocket (Fig. 5b). Notably, the base-recruiting mechanism is commonly observed in the ligand−G4 complex, which appears to be crucial for specific ligand recognition of DNA G4s[29,31,49,60−62]. Moreover, the similar binding orientation of berberine to *MYC*-G4 and *PDGFR-β* vacancy G4 has been previously reported [47,50].

Aside from the extensive stacking interaction between the A-berberine plane and outer G-tetrad core, potential electrostatic interactions can occur between the positively charged BerN7 and the negatively polarized carbonyl groups of the tetrad-guanine, analogous to a K$^+$ cation (Fig. 5b). It needs to be pointed out that additional intermolecular NOE cross-peaks between berberine and *KRAS*-G4 were

observed at high threshold levels, suggesting the coexistence of more than one berberine orientation at each binding pocket. For instance, the intermolecular NOE cross-peaks from BerMeB to G8H8 of the 5′-tetrad and from BerMeB to G6H8 of the 3′-tetrad were observed, respectively, revealing that the berberine orientation can be flipped for ca. 180° of the currently determined conformations (Fig. 5b). Whereas, insufficient NOE cross-peaks hindered the detailed structural determination of the minor species. Interestingly, the coexistence of distinct berberine orientations has been recently reported in a berberine-*MYC*-G4 complex [50].

## NMR solution structure determination of the 2:1 coptisine−*KRAS*−G4 complex

Similar experimental procedures were performed to obtain the solution structure of the 2:1 coptisine-*KRAS*-G4 binary complex. All imino, aromatic, and sugar resonances are assigned based on the NMR spectra compared with the berberine−*KRAS*−G4 complex (Supplementary Table 11). The H1-H1, H1-H8, H8-H8 connectivity patterns, NOESY sequential walk, and H6-C6/H8-C8 cross-peaks (Supplementary Figs. 22−26), as well as free and bound coptisine protons are presented

**Table 1 | NMR restraints and structural statistics for the free *KRAS*-G4 and its complex with berberine and coptisine**

| | *KRAS*-G4 | BER-*KRAS*-G4 | COP-*KRAS*-G4 |
|---|---|---|---|
| **NOE-Based Distance Restraints** | | | |
| Total | 745 | 728 | 686 |
| Intra-residue | 516 | 446 | 432 |
| Inter-residue | | | |
| Sequential | 156 | 185 | 163 |
| Long-range | 73 | 60 | 54 |
| Ligand-G4 | - | 37 | 37 |
| **Other restraints** | | | |
| Hydrogen bonds | 48 | 48 | 48 |
| Torsion angles | 24 | 24 | 24 |
| G-tetrad planarity | 48 | 48 | 48 |
| **Structural statistics** | | | |
| **Pairwise heavy atom RMSD (Å)** | | | |
| G-tetrad core | 0.58 ± 0.21 | 0.58 ± 0.17 | 0.49 ± 0.15 |
| All residues | 0.71 ± 0.21 | 0.75 ± 0.16 | 0.64 ± 0.14 |
| **Restraint violations (Å)** | | | |
| Max. NOE | 0.12 | 0.14 | 0.12 |
| Mean NOE | 0.001 ± 0.006 | 0.002 ± 0.010 | 0.001 ± 0.008 |

(Supplementary Figs. 24, 28, 30, and Supplementary Table 16). The same G-tetrad core was obtained compared to the free *KRAS*-G4 with G-tracts orientation parallel to each other (Figs. 1a and 4c). The larger Δ𝛿 values are shown for the flanking residues and the outer G-tetrad guanines (Supplementary Figs. 33, 34), suggesting that coptisine stacks above the two outer G-tetrads. NMR solution structures of the 2:1 coptisine-*KRAS*-G4 binary complex were thus solved by NOE-restrained MD simulations (Fig. 6, Supplementary Figs. 11 and 12, Table 1, and Supplementary Tables 11–15). The ten lowest-energy complex structures are presented in Fig. 4c and the RMSD values are shown in Table 1.

The overall structure of the coptisine–*KRAS*–G4 complex is almost identical to that of the berberine–*KRAS*–G4 complex (Figs. 4, 5, and Supplementary Figs. 11–13). A similar A-coptisine plane is formed on both the 5′-end and 3′-end capping structures, as well as potential electrostatic interactions between the positively charged CopN7 and the negatively polarized outer tetrad-guanine carbonyl groups. In both the 5′-end and 3′-end complexes, BerN7 and CopN7 are positioned over the central channel above the outer G-tetrad for potential electrostatic interactions. Notably, while the 3′-end sites of the two complexes are indistinguishable, the 5′-end sites differ as the A3:coptisine pair is stabilized by a potential H-bond, as supported by the observation of NOE cross-peaks between coptisine protons and A3H2 (Fig. 5c, Supplementary Table 12). By contrast, the proposed H-bond is weaker in the case of the A3:berberine pair, since the rotatable OCH₃ group is

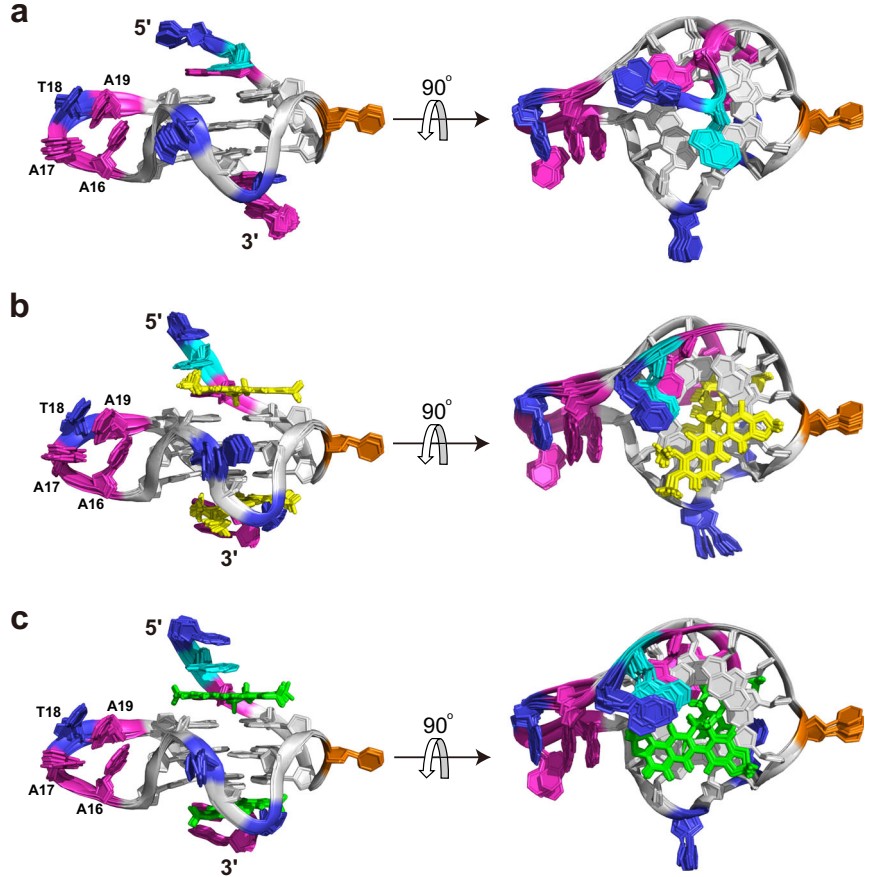

**Fig. 4 | NMR solution structures of *KRAS*-G4 and its complexes with berberine and coptisine.** Superposition of the ten lowest energy NMR structures of the free *KRAS*-G4 (**a**), berberine-*KRAS*-G4 (**b**), and coptisine-*KRAS*-G4 (**c**) by NOE-restrained structure calculations. Yellow, berberine; green, coptisine; cyan, flanking guanine; gray, tetrad guanine; magenta, adenine; blue, thymine; orange, cytosine.

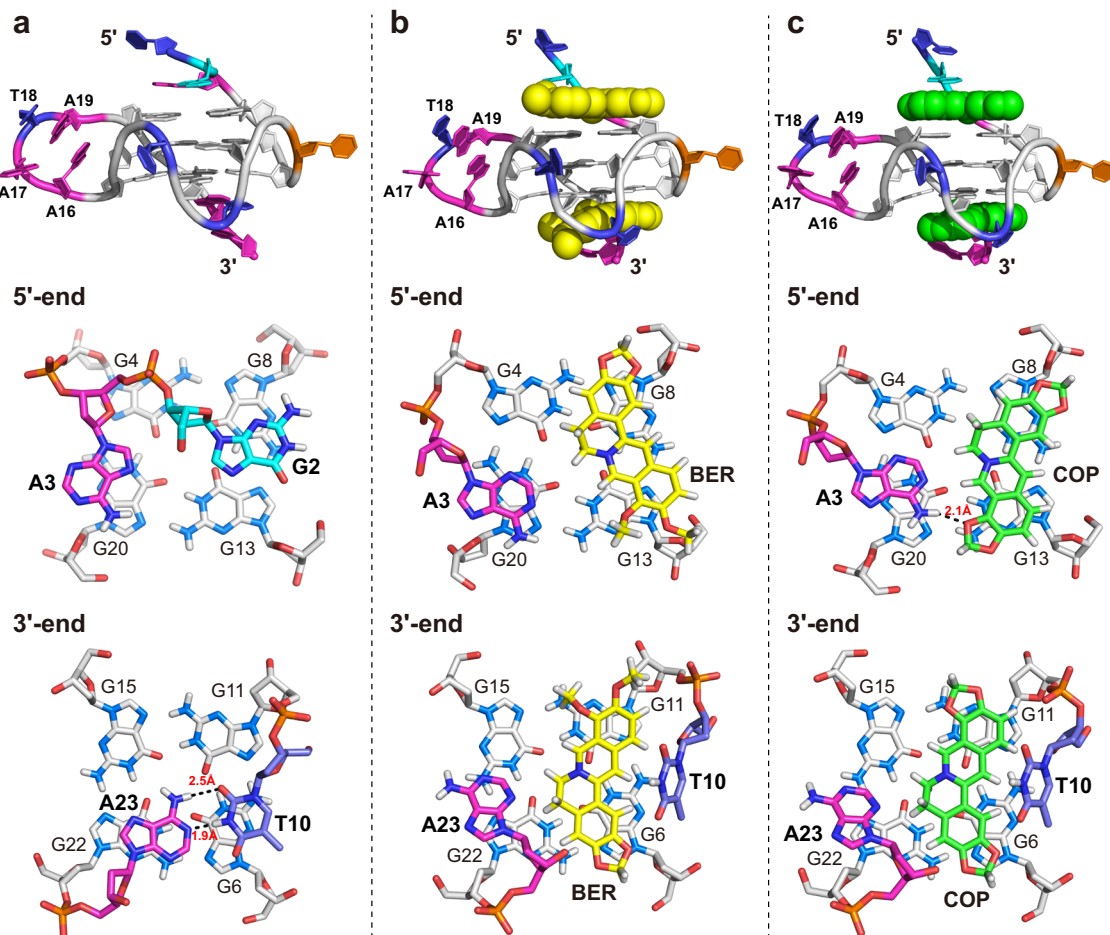

**Fig. 5 | Solution structure details of *KRAS*-G4 and its complexes with berberine and coptisine.** Cartoon representation and 5′-end and 3′-end top views of the free *KRAS*-G4 (**a**), the berberine-*KRAS*-G4 (**b**), and the coptisine-*KRAS*-G4 (**c**) complexes (Protein Data Bank IDs: 7X8N, 7X8M, and 7X8O). Yellow, berberine; green, coptisine; cyan, flanking guanine; gray, tetrad guanine; magenta, adenine; blue, thymine; orange, cytosine. Potential hydrogen bonds are shown as dashed lines.

not a suitable H-bound acceptor compared to the methylenedioxy five-member ring F (Fig. 5). A similar H-bond has been previously reported in the complex of epiberberine to a telomeric G4[49]. Moreover, solvent exchange experiments showed that the guanine imino protons in ligand-*KRAS*-G4 complex structures are significantly protected compared to free *KRAS*-G4, which could be attributed to the π-stacking, electrostatic, and/or H-bond interactions between the ligand and *KRAS*-G4 (Supplementary Fig. 35). Interestingly, the 4-nt loop structure is not involved in the binding pocket formation in both the berberine-Pu24m1 and coptisine-Pu24m1 complexes (Figs. 4 and 5).

**Berberine and coptisine stalled the replication of wild-type *KRAS*-G4 containing DNA fragment**
To investigate if berberine and coptisine can stabilize the *KRAS*-G4 in the extended DNA context, we performed a DNA polymerase stop assay using a DNA template that contains the wild-type *KRAS*-G4 forming motif. The formation of *KRAS*-G4 in the template DNA was confirmed by the 1D ¹H NMR experiment, as imino protons that characterized G4 structures were shown at 10–12 ppm (Supplementary Fig. 36). The G4 secondary structure formation in the template strand can block the *Taq* DNA polymerase from synthesizing the complementary strand DNA[49,63]. As shown in Fig. 7a, a premature product (stalled product) was observed faintly in the 50 mM K⁺-containing reactions, indicating *KRAS*-G4 formation. Notably, the addition of berberine and coptisine significantly blocked the Taq DNA polymerase in a dose-dependent manner, as shown by the increasing amount of the stalled product in the 50 mM K⁺-containing

solutions. The results suggest that berberine and coptisine may be able to stabilize the physiological relevant *KRAS*-G4 and therefore affect the function of *KRAS*-G4 in cells, such as replication and transcription regulation.

**Berberine and coptisine significantly lowered *KRAS* oncogene transcription levels in cancer cells**
The stabilization of G4 in the *KRAS* oncogene promoter was found to lower *KRAS* gene transcription levels[28,32,39]. To determine the effect of berberine and coptisine on the *KRAS* transcription levels, a quantitative reverse transcription PCR (qRT-PCR) experiment was performed on non-small cell lung cancer cells (H460 and A549) which harbor *KRAS* mutations with unfavorable therapeutic outcomes, and normal human bronchial epithelial cells (BEAS-2B)[3,64]. The IC₅₀ values of berberine and coptisine toward these three cell lines were firstly determined (Supplementary Fig. 37). Subsequently, H460, A549, and BEAS-2B cells were incubated with berberine and coptisine for 24 h and 48 h, respectively, the *KRAS* mRNA levels were then measured. The results showed that berberine and coptisine significantly lowered the *KRAS* mRNA levels in the two cancer cell lines, while not in the normal BEAS-2B cells (Fig. 7b and Supplementary Figs. 38 and 39). Furthermore, the colonies formation numbers were significantly decreased in the berberine and coptisine treatment A549 and H460 groups, but not in BEAS-2B cell lines (Supplementary Fig. 40). Collectively, these results indicated that coptisine and berberine could inhibit the *KRAS* oncogene transcription levels and the proliferation of the cancer cells, suggesting they are promising *KRAS*-G4 targeting drugs.

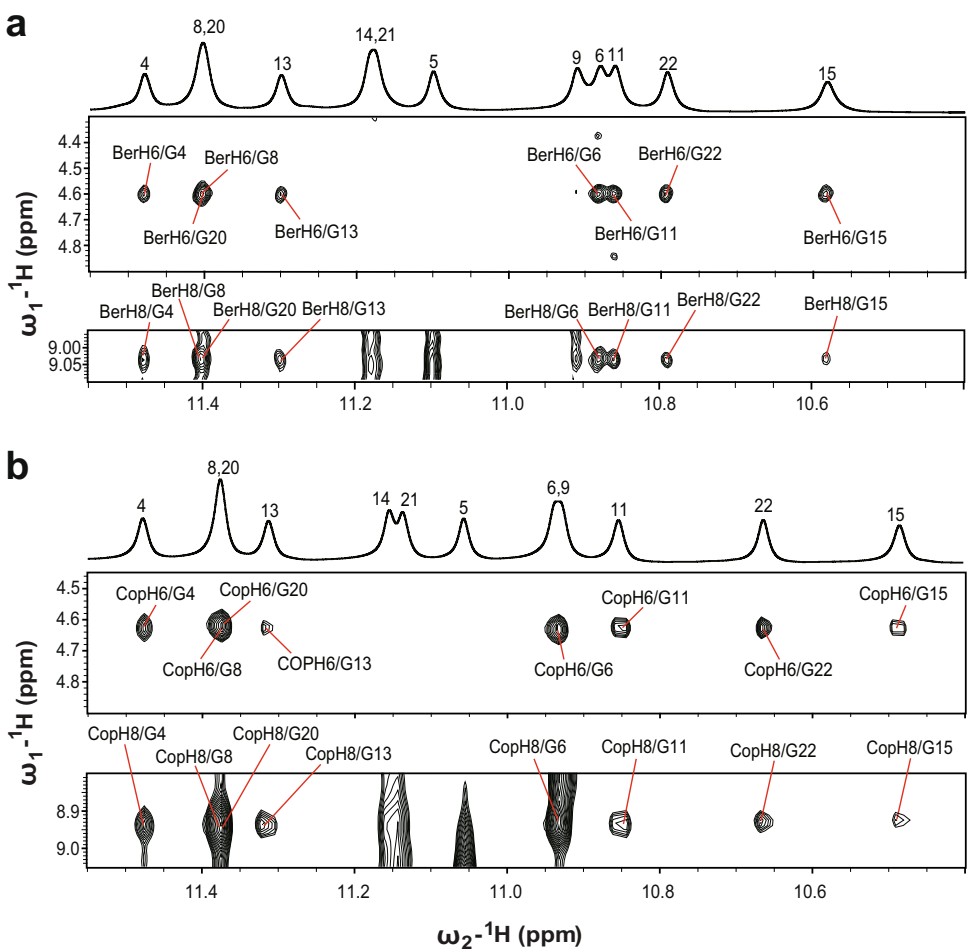

**Fig. 6 | 2D NMR spectra of *KRAS*-G4 in complex with berberine and coptisine.** Select regions of the 2D-NOESY spectra of 3:1 berberine·*KRAS*-G4 (**a**) and 2.5 :1 coptisine·*KRAS*-G4 (**b**) complexes in H$_2$O showing intermolecular cross-peaks between compound and DNA imino protons. Conditions: 1.5 mM Pu24m1 DNA, pH 7, 50 mM K$^+$ solution, 35 °C, DMSO-$d_6$ < 3.5%.

## Designing new berberine and coptisine derivatives with enhanced selectivity and/or affinity for *KRAS*-G4

Although berberine and coptisine show promising *KRAS*-G4-targeting therapeutic potential, there is space for chemical modifications to achieve enhanced selectivity and/or affinity (Figs. 1b, 8, and Supplementary Fig. 41). Firstly, the 9- and 10-OMe group of berberine or the methylenedioxy ring F of coptisine can be modified with a functional side chain for specific 4-nt long loop and groove interactions. Secondly, the methylenedioxy ring E of berberine and coptisine are the sites to introduce the second side chains for groove interactions. Lastly, but most importantly, the H1, H12, and H13 of berberine and coptisine toward the bulging thymine T10, which can be used to introduce specific H-bond interactions. Collectively, new berberine and coptisine derivatives can be designed by introducing distinct side chains or judicious chemical modifications to achieve enhanced selectivity and/or affinity for *KRAS*-G4.

## Discussion

Although several types of free *KRAS*-G4 structures and some *KRAS*-G4-interactive small molecules have been reported, to date, no available *KRAS*−G4−ligand complex structure has yet been determined, which seriously hinders the structure-based rational design of *KRAS*-G4 specific drugs. Herein, we determined the NMR solution structures of the bulge-containing *KRAS*-G4 bound to two natural isoquinoline alkaloids, berberine, and coptisine, respectively. The determined complex structure shows a 2:1 binding stoichiometry with each

berberine or coptisine recruits the adjacent flanking adenine residue to form a "quasi-triad plane" that stacks over the two external G-tetrads, which is similar to the recently determined complex structures of berberine bound to a *MYC*-G4 or a dGMP-fill-in *PDGFR-β* vacancy G4[29,47]. The binding involves both π-stacking and electrostatic interactions, and further enhancing affinity and selectivity of berberine and coptisine derivatives to the *KRAS*-G4 are proposed by introducing additional side chains or specific hydrogen bonds. Moreover, berberine and coptisine significantly stall the *Taq* DNA polymerase synthesis of DNAs and lower the *KRAS* mRNA levels in cancer cells. Notably, target specificity of the two compounds appears to be weak and an in-depth structure−activity−relationship (SAR) analysis based on the determined high-resolution complex structures is highly required. Collectively, our study contributes structural insights into the ligand interactions with *KRAS*-G4 and provides a model system for the design of specific *KRAS*-G4-interactive small molecules.

## Methods
### Sample preparation

Labeled and unlabeled DNA oligonucleotides were obtained from Sangon Biotech (Shanghai, China) Co., Ltd. The DNA was dissolved in a buffer containing 37.5 mM KCl, 12.5 mM K$_2$HPO$_4$/KH$_2$PO$_4$, pH 7, 10/90% D$_2$O/H$_2$O. The DNA concentrations were quantified by a UV spectrometer using the extinction coefficients. Berberine and coptisine were purchased from Shanghai Standard Technology Co., Ltd., which were dissolved in DMSO-$d_6$ to a stock solution of 40 mM.

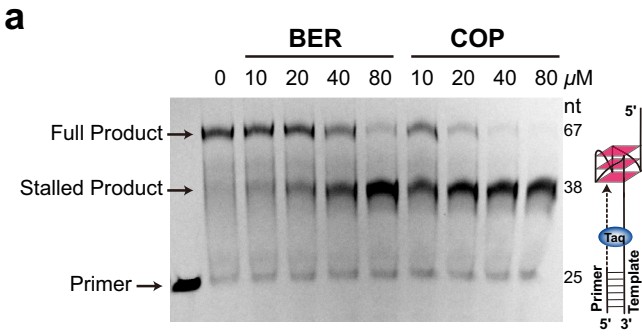

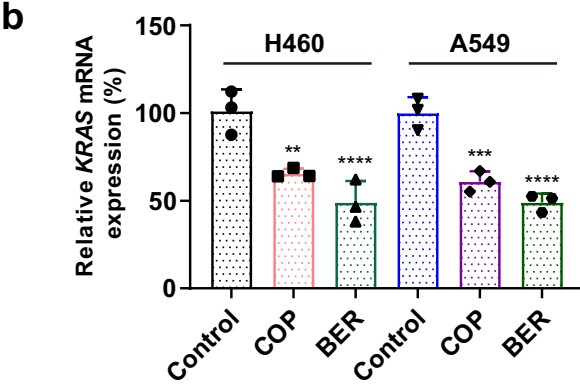

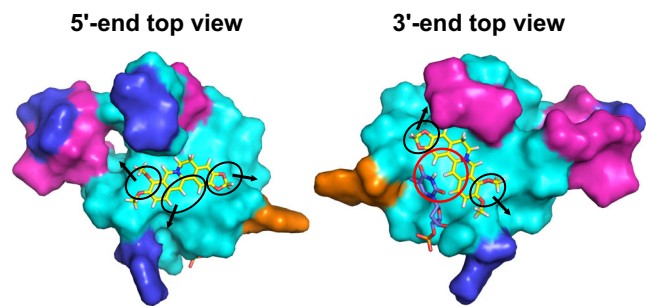

**Fig. 8 | Suggested modifications of berberine enabling additional interaction with the *KRAS*-G4.** The black circles indicate the positions to be modified by introducing side chains for groove interactions. The black arrows show the grooves at 5′- and 3′-sites where the attached side chain will locate. The red circle suggests the positions to introduce hydrogen bonds between berberine and thymine T10. The *KRAS*-G4 is shown in cartoon representation. Cyan, guanine; magenta, adenine; blue, thymine; orange, cytosine; yellow, berberine.

**Fig. 7 | Functional characterization of *KRAS*-G4. a** Taq DNA polymerase stop assay using a DNA template containing a wt human *KRAS*-G4 forming sequence shows that berberine and coptisine can stabilize the *KRAS*-G4 for replication inhibition. **b** qRT-PCR results show that berberine (24 μM) and coptisine (10 μM) lower *KRAS* mRNA levels in both H460 and A549 cancer cells for 24 h. DMSO (<0.1 %) was used as the negative control (no inhibition, 100%). The relative *KRAS* mRNA levels were normalized with *GAPDH*. The experiments were run in triplicate. Data are presented as mean values ± SD. *P* values (**$P < 0.01$, ***$P < 0.001$, ****$P < 0.0001$) were determined by one-way ANOVA with post hoc Dunnett, relative to DMSO control.

### Nuclear magnetic resonance (NMR) spectroscopy experiments

1D and 2D NMR data were collected on a Bruker AV-600 spectrometer (QCI cryoprobe). The w5 water suppression was used. 2D NOESY were collected at temperatures of 20, 25, and 35 °C with mixing times of 80, 150, 300, and 350 ms in 90% $H_2O$/10% $D_2O$ solutions. Moreover, some representative NOE build-up curves have been determined (Supplementary Fig. 42). The NOE distance calibration was done using the cytosine H5/H6 distance (2.4 Å). DQF-COSY spectrum was collected at 25 °C. $^1H$-$^{13}C$ HSQC spectra were collected using the hsqcetgpsi pulse sequence with $^1J_{(C, H)} = 145$ Hz. Chemical shift calibration was done indirectly for $^{13}C$ relative to DSS and directly for $^1H$ according to the water signal relative to DSS. The NMR data were processed and analyzed with Topspin 4.1.1 (Bruker) and Sparky (UCSF), respectively. For 2D NOESY spectra, 480–580 increments were accumulated in the indirect (F1) dimension and 2048 points in the direct (F2) dimension. The spectral widths were 16.5 ppm and 10 ppm for F2 and F1 dimensions, respectively, or 16.5 ppm for both dimensions. The processing parameters were 1024 points for F1 and 2048 points for F2, with a Sine bell shift (SSB) of 3 in both dimensions and the QSINE window function.

### NOE-restrained simulated-annealing

NOE-restrained simulated-annealing was performed as previously established[15,29,47,65]. NOESY cross-peaks were defined as very weak (6.0 ± 1.5 Å), weak (5.5 ± 1.5 Å), medium (4.0 ± 1.5 Å), and strong (2.9 ± 1.1 Å) according to the spectra collected with mixing times of 80,

150, and 350 ms. For instance, the intra-residue NOE cross-peaks of sugar H1′/H2′, H1′/H2″, H1′/H4′, thymine H6/HMe, and cytosine H5/H6 are defined as strong NOEs; the intra-residue NOE cross-peaks of sugar H1′/H3′, H4′/H2′, H4′/H2″, and anti-conformation H1′/H8 are defined as medium NOEs. The other NOE cross-peaks are classed based on intensity that compared to these well-define NOEs. Notably, one repulsive restraint of 7.5 ± 1.5 Å was applied to prevent the flanking residue T1 from a position close to the 5′-end tetrad not in line with experimental data for the berberine−DNA and coptisine−DNA complexes, respectively. Exchangeable protons were defined as very weak (6.0 ± 1.2 Å), weak (5.0 ± 1.2 Å), and medium (4.0 ± 1.2 Å). Berberine−DNA and coptisine−DNA intermolecular cross-peaks were defined as very weak (6.0 ± 1.5 Å), weak (5.0 ± 1.5 Å), and medium (4.0 ± 1.5 Å). Overlapped and ambiguous resonances were defined as a distance of 5.0 ± 2.0 Å. Dihedral restraints of 170°–310° and 200°–280° for *anti*-conformations in the loop and within the G-tetrad, respectively, were used. Xplor-NIH 2.48 simulated annealing protocol was performed to obtain 100 initial structures[66]. Berberine and coptisine parameter files were obtained from ChemDraw 18.2 and further optimized and calculated with the Gaussian09 program[67]. The sander module of Amber 20 was used for simulated annealing in implicit water of the 100 initial structures[68]. The OL15 DNA force field was employed with modifications of the parmbsc0[69–71]. The force constants of 50, 30, and 20 kcal mol$^{-1}$ Å$^{-2}$ for hydrogen bond, G-tetrad planarity, and NOE-based distance restraints were applied, respectively. Moreover, 200 kcal mol$^{-1}$ rad$^{-1}$ of glycosidic angle restraints were used. 100 structures were firstly equilibrated at 100 K for 5 ps and then heated to 1000 K for 10 ps. The system was held at 1000 K for 30 ps and subsequently cooled down to 100 K by 45 ps. In the end, 0 K was achieved in the last 10 ps and 20 lowest-energy structures were obtained.

### NOE-restrained molecular dynamics refinement

NOE-restrained molecular dynamics refinement was carried out as previously described[15,29,47,65]. The 20 lowest-energy structures from the simulated-annealing process were solvated with TIP3P water and neutralized by $K^+$ cations, including two $K^+$ in between the three G-tetrads. The system was then equilibrated and minimized by fixing the DNA position with 25 kcal mol$^{-1}$ Å$^{-2}$ force constants. Subsequently, the system was heated in 20 ps from 100 to 300 K with constant volume, while the DNA force constant was decreased gradually from 5, 4, 3, 2, 1 to 0.5 kcal mol$^{-1}$ Å$^{-2}$ in each 10 ps. The final 4 ns production run was conducted in the Amber pmemd module with constant pressure and snapshots were taken at each 1 ps. Force constants of 25, 10, and 5 kcal mol$^{-1}$ Å$^{-2}$ for hydrogen bond, NOE-based distance, and G-tetrad

planarity restraints were employed, respectively (Without G-tetrad planarity restraints, similar final structures were obtained, Supplementary Fig. 43). Finally, the last 500 ps of the trajectories were averaged and energy-minimized for 500 steps in the vacuum after removal of the water molecules and cations. Final ensemble and deposition were conducted using the ten lowest energy structures. The VMD and PyMOL software were used for analysis and visualization [72,73].

## Circular Dichroism (CD) experiments

CD data were collected on a Jasco-1500 spectropolarimeter (Jasco Inc., Japan). 20 μM DNA samples were prepared in a buffer containing 3.5 mM $K_2HPO_4/KH_2PO_4$, 11.5 mM KCl, pH 7. The DNA samples were annealed before use. Berberine or coptisine were added in the desired concentrations. CD spectra were collected using a 1 mm path length quartz cuvette at 25 °C. The blank correction was applied. For CD melting, the sample was heated from 25 to 95 °C with a heating rate of 2 °C/min, and the CD ellipticity at 264 nm was recorded. The melting temperature was then obtained at the intersection between the median of the fitted baselines and the melting curve.

## Fluorescence measurements

Fluorescence data were collected on a Jasco-FP8300 spectrofluorometer (Jasco Inc., Japan). Fluorescence spectra were acquired in a 1 cm path length quartz cell for the emission spectra from 520 to 600 nm. The excitation wavelength was set to 377 nm. The berberine or coptisine concentrations were fixed to 0.2 μM in a 50 mM $K^+$-containing solution. Pu24m1 DNA at the desired concentration was added. Fluorescence spectra were collected after 2 min incubation at each time. The $K_d$ value was obtained by fitting the data to an equation using GraphPad Prism software, with a 2:1 binding stoichiometry: $F = F_{min} + (F_{max} - F_{min})[(2D_T + C_T + K_d) - [((2D_T + C_T + K_d)^2 - (8D_T C_T))^{1/2}]/(2C_T)$. $F$, ligand-induced fluorescence intensity; $C_T$, ligand concentration; $D_T$, complex concentration.

## DNA polymerase stop assay

This assay was performed as previously described[49,74]. The 5′-end FAM labeled primer (5′-FAM-TAATACGACTCACTATAGCAATTGC) was mixed with template DNA in a 1.2:1 equivalent (5′-TGAATCCT-GAGGGCGGTGTGGGAAGAGGGAAGATAGCTGCACGCAATTGCTA-TAGTGAGTCGTATTA-3′). The mixtures were annealed by heating to 95 °C for 5 min then cooling to room temperature. Berberine and coptisine were added at various concentrations and incubated at room temperature for 3 hours. Primer extension was performed for 30 min in a 50 μL reaction buffer containing 0.2 μM DNA mixtures, 0.1 mM dNTP, 1.25 U/μL Thermo *Taq* DNA polymerase, 50 mM $K^+$ (pH 7.0), and 2 mM $MgCl_2$. The DNA products were resolved on a 12% denaturing polyacrylamide gel. DNA fragments were visualized by scanning on a ChemiDOC XRS + system (BIO-RAD, USA) and processed by Image Lab 6.1.

## Native electrophoretic mobility shift assay (EMSA)

EMSA gel data were collected using a 1.5 mm thick 10 × 7 cm native gel, which contained 12.5 mM KCl and 16% acrylamide, pH 8.0. DNA samples, in the absence and presence of berberine and coptisine, were prepared in a 37.5 mM KCl, 12.5 mM phosphate buffer, pH 7.0. DNA bands were visualized at 260 nm under UV light.

## Cell viability assay

Cells were grown in the RPMI 1640 medium supplemented with 10% FBS (Gibco, USA) and 1% Penicillin–Streptomycin Solution at 37 °C in a humidified atmosphere of 95% air and 5% $CO_2$. Cells were seeded in a 96-well plate at a concentration of $1.0-2.0 \times 10^3$ cells/well and treated with different gradient concentrations of coptisine and berberine maintained at 37 °C for 48 or 72 h. The cells were then treated with WST-8 [2-(2-methoxy-4-nitrophenyl)-3-(4-nitrophenyl)-5-(2,4-disulfophenyl)-2H-tetrazolium, monosodium salt] (MedChemExpress, Monmouth Junction, NJ, USA) for 3 h at 37 °C. The absorbance at 450 nm was determined with a microplate reader. The $IC_{50}$ values were determined from the sigmoidal dose–response curves using GraphPad Prism.

## Colony formation assay

Cells were seeded in 6-well plates containing a complete growth medium overnight and then treated with DMSO (<1%), 10 μM coptisine, or 24 μM berberine for 24 h. After 24 h, the medium was removed and replaced with fresh medium for 7–8 days. The cells were then washed with PBS and fixed with a 4% paraformaldehyde fix solution (Servicebio, China) for 10 min. Subsequently, the cells were stained with crystal violet staining solution (Beyotime, China) for 10 min. Finally, the plates were washed, dried, and photographed.

## Quantitative reverse transcription PCR (qRT-PCR)

Total RNA was collected using the RNA extraction kit (Yishan Bio., China) according to the manufacturer's instructions. RNA was reverse transcribed using the HiScript Q RT SuperMix kit (Vazyme, China). qRT-PCR was performed using Lightcycler 480 (Roche, Germany). SYBR Green Real-Time PCR Master Mix (Vazyme, China) was selected as the amplification reagent. *GAPDH* was used as an endogenous control. $\Delta\Delta C_t$ method was used to analyze the mRNA levels and each experiment was performed in independent triplicate. The

*KRAS* primer. F: 5′-CAGTAGACACAAAACAGGCTCAG-3′
R: 5′-TGTCGGATCTCCCTCACCAATG -3′

*GAPDH* primer. F: 5′-GGTGAAGGTCGGAGTCAACGG-3′
R: 5′-GAGGTCAATGAAGGGGTCATTG-3′

## Statistical analysis

All experiments were run in 2–3 independent replicates. Statistical significance of differences between groups was analyzed using one-way ANOVA with post hoc Dunnett, relative to the control (**$P < 0.01$, ***$P < 0.001$, ****$P < 0.0001$).

## Reporting summary

Further information on research design is available in the Nature Research Reporting Summary linked to this article.

## Data availability

Source data are provided in this paper. The data that support the findings of this study are available from the corresponding authors upon reasonable request. The coordinates and experimental details generated in this study have been deposited in the Protein Data bank under accession codes 7X8N (free *KRAS*-G4), 7X8M (2:1 berberine–*KRAS*–G4 complexes), and 7X8O (2:1 coptisine-*KRAS*-G4 complexes).

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

## Acknowledgements

This research was supported by the National Natural Science Foundation of China (K.W., 81773886 and L.K., 82173707), the Program for Jiangsu Province Innovative Research Scholar (K.W., 1402100013), the Natural Science Foundation of Jiangsu Province (K.W., BK20210420), the Scientific Research Foundation of Nanjing for Returned Chinese Scholar (K.W., No. 1412100026), and the Scientific Research Foundation for high-level faculty, China Pharmaceutical University, Nanjing, China (K.W., No. 3150020065). We thank Prof. Dr. Danzhou Yang and Dr. Jonathan Dickerhoff at Purdue University (USA) for their comments and for proofreading the manuscript.

## Author contributions

K.W. and L.K. conceptualized and designed the experiments. K.W., Y.X., T.Y., and M.Y. supervised the study. K.W. and Y.L. performed most of the experiments with J.L., C.X., Y.W., W.G., and Y.L. as assistants. K.W., Y.L., and L.K. wrote the manuscript. All authors have read and approved the final version of the manuscript.

## Competing interests

The authors declare no Competing interests.
