## [Peer Review File · Nature Communications]

Structural Insight into the Bulge-containing KRAS Oncogene Promoter G-Quadruplex Bound to Berberine and CoptisineREVIEWER COMMENTS

Reviewer #1 (Remarks to the Author):

To the reviewers: Kong and co-workers provide a solid piece of work with robust data. The analysis, including structural work, of the results seems correct and the discussion large and deep enough to arrive to the authors conclusions. The decision to extend the KRAS short G-quadruplex sequence from what other authors published in previous studies was a good one. The results are well supported by a series of NMR spectra, although we cannot inspect them correctly because they are rather incomplete in size. Refinement should be clearly stated in methods. Nevertheless, the lack of some supporting experiments seems necessary to exhaustively support the claims. Some are described below.

General comments:

1. The choice of berberin and coptisine as ligands for G4 in KRAS seems interesting, nevertheless a radar plot (ex: DOI: 10.1002/chem.20160488) should be provided where these ligands are shown against other G4 models such as H-tel and c-myc.
2. Please indicate how much DMS you have in each sample that gave the NMR spectra in all figures. If contains less than 1% just mention residual DMSO.
3. Please provide in SI, a full 1D NMR spectra of oligos without and with each ligand (2 molar ratio).
4. Provide the acquisition data points and the processing details, especially the zero-filling points.
5. Provide in SI an entire 2D NOESY spectra without and with both ligands. Otherwise deposited them at the BMRB and insert the codes.
6. The Taq DNA polymerase stop assay should be done also with normal (non-cancerous) cell lines as reference for cell-line specificity and disease interest.
7. Cellular time-testing experiments are short. Normal practice would be at least 7-15 days.
8. PAGE 12: " that additional intermolecular NOE cross-peaks between berberine and KRAS-G4 were observed at high threshold levels, suggesting the coexistence of more than one berberine orientation at each binding pocket"
9. This is an interesting observation; could the authors distinguish what are the non-preferential orientations? It is a difficult question, but sometimes necessary to try to improve the ligands to fit more selectively and with tighter binding.

Reviewer #2 (Remarks to the Author):

The manuscript by Wang et al. examines the NMR solution-state structures of KRAS -G4 as a molecular basis for elucidating the specific recognition of the thymidine bulge by two natural isoquinoline alkaloids, berberine and coptisine. The determined structures of the complexes show a 2:1 binding stoichiometry with both berberine and coptisine stacked on the two outer G-tetrads and forming a quasi-triad plane with adenine.

The manuscript is written clearly and builds on the earlier work of the author group and others on alkaloids such as berberine. In this regard, some praise is overemphasised. This study provides molecular details of ligand interactions with G4, but does not 'contribute a new platform for cancer therapy' as stated in the conclusions. Nevertheless, the results are interesting and innovative. Some challenges of the study and minor modifications are suggested below.

On p. 8 authors state that 'a much smaller T_m change was shown from 2 to 4 equivalents of drug addition, suggesting two dominant binding sites of berberine and coptisine to KRAS-G4.' This statement needs more detailed elaboration as it appears that at 2 equivalents both binding sites are saturated completely. As no changes are observed with NMR above 2 eqv. of alkaloids according to Figure 1C, what molecular interpretation is offered for further thermal stabilization, which is in fact considerable?

Fluorescence-based binding assay shows that cca. 10 eqv. excess of 'Pu24m1 DNA' was needed to reach a saturated state. How were exp. data fitted to calculate binding affinities of two isoquinoline alkaloids to the KRAS-G4? Using 1:2 or 1:4 model?
Can relatively higher ΔF values for COP (Fig. 2b) be used to infer different binding modes of the two alkaloids?

Assignment of KRAS-G4 spectra with the aid of the reported 22R DNA assignment (ref. 40) may be problematic. A simple comparison shows that imino regions of Pu24m1 and 22R differ considerably. This needs to be elaborated and explained clearly!

As per statement 'A16-A19-T18 base stacking organization, which is connected by an A17 linker' on p. 10 of the main text, numbering of selected residues in Fig. 4 may be helpful to assist a potential reader.

Statement 'that berberine and coptisine can stabilize the physiological relevant KRAS-G4 in the human genome DNA context' needs to be rephrased as it represents interpretation of intensity of bands on a PAGE rather than actual KRAS-G4 formation in cells.

DMSO is used as a negative control in qRT-PCR and supposedly gives no inhibition. However, addition of DMSO could disrupt (reduce) KRAS-G4 formation. It is not clear from results in the current version of manuscript if authors have considered such an effect on destabilization of G4 and subsequent functional consequences. How significant is partial unfolding of KRAS-G4 in the presence of DMSO and how does it affect relative KRAS mRNA expression levels?

Why was berberine (and coptisine) dissolved in DMSO when solubility in water can achieve over 5 mM concentration at room temperature? DNA concentration was 0.15 mM (Fig. 1c) which should make it possible to perform titration with NMR in completely aqueous solution.

Minor.

- Footnotes of Fig. 2 need to be expanded to provide sufficient details.
- Somewhat surprisingly 'various 2D NMR spectra at 3:1 berberine-DNA ratio were collected', but respective spectrum in Figure 6 was taken at 2:1 ratio. Is this a typo?
- Authors may wish to modify Fig. 8 to better illustrate introduction of additional side chains with increased resolution of suggested interactions with grooves.
- Some grammatical corrections may be needed throughout the manuscript. As an example, the first sentence of Discussion section on p. 17 needs to be finished or connected with the next one, removing full stop in between.
- Are G-tetrad planarity restraints really needed? An attempt should be made to release them, at least in the final stages of SA calculations. And report the results, whatever they show.

Reviewer #3 (Remarks to the Author):

Wang et al report a structural analysis of the interaction of a G-quadruplex DNA formed in the promoter region of the KRAS oncogene with the small molecules berberine and coptisine. They use NMR, CD and EMSAs to characterize the binding and then use the KRAS1234 G4 with flanking regions for NMR structure determination using ^1H NMR spectra only. Functional effects of the two small molecules are shown by DNA polymerase assays in the absence and presence of inhibitors using a DNA template with the KRAS G4 motif and qRT-PCR monitored effects of transcription in lung cancer cells. The authors then discuss potential modification of the small molecules to improved binding and inhibition, however, not experimental verification is done. The manuscript is interesting and studies a biomedically important topic. Identifying and optimizing specific inhibitors of DNA G4 quadruplexes is interesting, but also challenging. The presented structural work contributes to this. The work appears technical sound, although some more

experimental details and additional experimental support should be provided. The activity in cells seems to be consistent with the proposed inhibition, but given the relatively weak binding affinity there is concern how specific the compounds indeed can function in cells.

To support the structural analysis the structure-based optimization of the small molecules is discussed only. However, this should be performed experimentally and validated by NMR and the functional assay to support the conclusions made and increase the impact of the work.

NMR chemical shifts, restraints and coordinate files must be deposited in BMRB and PDB and accession codes provided in the manuscript.

Specific comments;

- Please mention K_d determined in the text (not only referring to Fig. 2c)
- The stoichiometry of the complexes seems not fully clear, saturation in the NMR titrations is mentioned at 3:1, but the CD seems to indicate 2:1? How do the authors rationalize the additional spectral change of imino signals from 2:1 to 3:1 (Fig. 1c), could this not indicate additional contacts and/or conformational changes beyond the 2:1 complex?
- Likewise: the melting points increase significantly from 2 to 4 eq (Fig. 2c), how is this explained?
- p.12: the authors speculate that competition between an A-T basepair and the A-berberine plane may explain that an unstable 3'-end complex is formed at 2:1 drug ratio. This is not clear, please explain and provide some experimental support.
- Can the authors provide some experimental support, chemical shift changes, solvent exchange rate differences, J-couplings for the proposed charged and/or H-bond interactions?
- How were NOEs calibrated, how was spin diffusion considered?
- The ligand signals and intermolecular NOEs are severely line-broadened (Suppl Fig. 8, 11), - how could the authors analyze and integrate these NOEs for structural analysis.
- Please provide 2D NMR spectra superpositions showing the differences in chemical shifts of small molecules free and bound to the G4.
- How were torsion-angle restraints derived from which experimental data?
- The DNA polymerase stalling occurs only above 40 μM berberine concentration, how is this consistent with the binding affinity of 0.5 μM ?
- Can the authors provide evidence that the G4 is indeed formed in the template for the DNA polymerase assay, and if possible also in the cell-based functional assay?
- English language should please be checked, some phrases/wording sounds awkward (i.e. "hotly" line 9 of the introduction; "Lots of intramolecular NOEs", p.11, ...)

Responses to reviewers' comments

Reviewer #1 (Remarks to the Author):

To the reviewers: Kong and co-workers provide a solid piece of work with robust data. The analysis, including structural work, of the results seems correct and the discussion large and deep enough to arrive to the authors conclusions. The decision to extend the KRAS short G-quadruplex sequence from what other authors published in previous studies was a good one. The results are well supported by a series of NMR spectra, although we cannot inspect them correctly because they are rather incomplete in size. Refinement should be clearly stated in methods. Nevertheless, the lack of some supporting experiments seems necessary to exhaustively support the claims. Some are described below.

Response: We appreciate the reviewer for these insightful comments. We have included more NMR spectra in the revised manuscript (Supplementary Figs. 14, 15, 18, 19, 23, and 24) and deposited the experimental details at the Protein Data Bank and BRMB ((7X8N, 36476, free *KRAS*-G4), (7X8M, 36475, berberine-*KRAS*-G4), and (7X8O, 36477, coptisine-*KRAS*-G4) (Page 26)). Refinement has been clearly described in methods (Pages 21-22). More updates can be seen as follows.

General comments:

1. The choice of berberine and coptisine as ligands for G4 in *KRAS* seems interesting, nevertheless a radar plot (ex: DOI: 10.1002/chem.20160488) should be provided where these ligands are shown against other G4 models such as H-tel and c-myc.

Response: We appreciate the reviewer's comments. We collected the new CD and NMR titration data for berberine and coptisine to *h-TEL*, *c-MYC*, and *VEGF* G4s, respectively (Supplementary Figs. 5-10). The radar plot is provided in Supplementary Fig. 6.

We discussed the obtained results in Pages 8-9. "To gain insight into the structural dependencies of berberine and coptisine binding, CD and NMR titration experiments were collected of these two compounds to *MYC*-G4 (parallel)⁴⁶, *VEGF*-G4 (parallel)⁵⁷, and human telomeric G4s (*Tel*-hybrid1 and *Tel*-hybrid2 G4s)^{31,49}. The structural selectivity profiles were evaluated by the ΔT_m values⁵⁸ obtained from the CD melting experiments. The data showed that the two compounds have preferences for the parallel G4s with a higher stabilization effect compared to the telomeric hybrid G4s (Supplementary Figs. 5 and 6). The results were consistence with NMR titration data because berberine and coptisine showed more specific binding to *KRAS*-G4, *MYC*-G4, and *VEGF*-G4, where a well-defined complex was formed at the ligand ratio of 2 or 3, with 12 new emerging imino proton peaks (Fig. 1 and Supplementary Figs. 7 and 8). In contrast, the compounds did not show specific binding to telomeric G4s as the ligand-G4 complex structures did not show 12 well-defined imino proton peaks (Supplementary Figs. 9 and 10)"

2. Please indicate how much DMSO you have in each sample that gave the NMR spectra in all figures. If contains less than 1% just mention residual DMSO.

Response: The comment is accepted. We included the percentage of DMSO in each relevant NMR spectra.

3. Please provide in SI, a full 1D NMR spectra of oligos without and with each ligand (2 molar ratio).

Response: The comment is accepted. We provided full 1D NMR spectra of 3:1 and 2.5:1 of berberine and coptisine to KRAS-G4, respectively, as well as the free KRAS-G4 in Supplementary Fig. 14, because the 2D spectra were collected at these conditions which showed well-resolved complex structures.

4. Provide the acquisition data points and the processing details, especially the zero-filling points.

Response: The comment is accepted. We included the acquisition data points and the processing details in methods (Page 20).

5. Provide in SI an entire 2D NOESY spectra without and with both ligands. Otherwise deposited them at the BMRB and insert the codes.

Response: The comment is accepted. We included the suggested 2D NOESY spectra in Supplementary Figs. 15, 18, and 23.

6. The Taq DNA polymerase stop assay should be done also with normal (non-cancerous) cell lines as reference for cell-line specificity and disease interest.

Response: As we described in methods (Page 23), the *Taq* DNA polymerase stop assay was conducted in a cell-free condition, which used Thermo *Taq* DNA polymerase. However, to confirm the cell-line specificity and disease interest, we tested the cytotoxicity of the two compounds to the normal human lung epithelial cells (BEAS-2B). It turned out that berberine and coptisine do not show significant cytotoxicity to the BEAS-2B cells (Supplementary Figs. 37,39, and 40).

7. Cellular time-testing experiments are short. Normal practice would be at least 7-15 days.

Response: The comment is accepted. We included the colony formation assay in Pages 17, 25 and supplementary Fig. 40. The colonies formation numbers were significantly decreased in the berberine and coptisine treatment A549 and H460 groups, but not in

BEAS-2B cell lines for 7-8 days.

8. PAGE 12: “that additional intermolecular NOE cross-peaks between berberine and KRAS-G4 were observed at high threshold levels, suggesting the coexistence of more than one berberine orientation at each binding pocket”

9. This is an interesting observation; could the authors distinguish what are the non-preferential orientations? It is a difficult question, but sometimes necessary to try to improve the ligands to fit more selectively and with tighter binding.

Response: Thanks for the comments. We observed the intermolecular NOE cross-peaks from BerMeB to G8H8 of the 5'-tetrad and from BerMeB to G6H8 of the 3'-tetrad, respectively, which indicated that the berberine orientation can be flipped for ca. 180° of the current determined conformations. Interestingly, the coexistence of several distinct berberine orientations have been recently reported in a berberine-MYC-G4 complex (*J. Med. Chem.* 2021, 64, 16205). We have included these discussions in the main text (Page 14).

Reviewer #2 (Remarks to the Author):

The manuscript by Wang et al. examines the NMR solution-state structures of KRAS -G4 as a molecular basis for elucidating the specific recognition of the thymidine bulge by two natural isoquinoline alkaloids, berberine and coptisine. The determined structures of the complexes show a 2:1 binding stoichiometry with both berberine and coptisine stacked on the two outer G-tetrads and forming a quasi-triad plane with adenine.

The manuscript is written clearly and builds on the earlier work of the author group and others on alkaloids such as berberine. In this regard, some praise is overemphasised. This study provides molecular details of ligand interactions with G4, but does not 'contribute a new platform for cancer therapy' as stated in the conclusions. Nevertheless, the results are interesting and innovative. Some challenges of the study and minor modifications are suggested below.

Response: We appreciate the reviewer for these insightful comments. We have revised the relevant text (Pages 2 and 19): “Our study provides molecular details of ligand interactions with KRAS-G4...” and “Our study thus contributes structural insight into the ligand interactions with KRAS-G4...”.

On p. 8 authors state that ‘a much smaller T_m change was shown from 2 to 4 equivalents of drug addition, suggesting two dominant binding sites of berberine and coptisine to KRAS-G4.’ This statement needs more detailed elaboration as it appears that at 2 equivalents both binding sites are saturated completely. As no changes are observed with NMR above 2 eqv. of alkaloids according to Figure 1C, what molecular interpretation is offered for further thermal stabilization, which is in fact considerable?

Response: Thanks for the comments. Yes, you are right. In Figure 1c, only minor changes were observed for the imino protons from 2 to 3 drug equivalents, where G15-H1 is clearly seen at 3 drug equivalents. For ligand binding at intermediate-to-fast exchange on the NMR timescale, a ligand:DNA ratio higher than its binding stoichiometry is often needed to push the binding equilibrium toward the formation of a well-resolved complex, which shows 12 sharp imino proton peaks (*J. Med. Chem.* 2021, 64, 16205; *J. Am. Chem. Soc.* 2019, 141, 11059; *Angew. Chem. Int. Ed.* 2016, 55, 12508). As large percentage of stable 2:1 binding complex structures were formed at higher ligand ratio, which is offered for further thermal stabilization. We added these details in Page 8.

Fluorescence-based binding assay shows that cca. 10 eqv. excess of 'Pu24m1 DNA' was needed to reach a saturated state. How were exp. data fitted to calculate binding affinities of two isoquinoline alkaloids to the KRAS-G4? Using 1:2 or 1:4 model? Can relatively higher deltaF values for COP (Fig. 2b) be used to infer different binding modes of the two alkaloids?

Response: Thanks for the comments. We used 2:1 (ligand:DNA) model to fit the obtained exp. data with an equation: $F = F_{\min} + (F_{\max} - F_{\min}) [(2D_T + C_T + K_d) - \sqrt{((2D_T + C_T + K_d)^2 - (8D_T C_T))^{1/2}}] / (2C_T)$. F, ligand induced fluorescence intensity; C_T, ligand concentration; D_T, complex concentration. We added these details in Page 23.

We reasoned that the relatively higher deltaF values for COP (Fig. 2b) is caused by the potential H-bond of the A3:coptisine pair. The 3'-end sites of two complexes are indistinguishable, but the 5'-end sites are different as discussed in page 15 (Fig. 5). in the main text.

Assignment of KRAS-G4 spectra with the aid of the reported 22R DNA assignment (ref. 40) may be problematic. A simple comparison shows that imino regions of Pu24m1 and 22R differ considerably. This needs to be elaborated and explained clearly!

Response: Thanks for the comments. We revised the sentence in Page 10. The assignment mainly follows standard strategies as described in our previous lab's published protocol (In *G-Quadruplex Nucleic Acids: Methods and Protocols*; Yang, D., Lin, C., Eds.; *Methods in Molecular Biology*; Springer: New York, NY, 2019, pp 157–176).

The reported 22R DNA assignment was referenced as both sequences have the same 3'-end flanking residues. To be specific, G22-H1 in free Pu24m1 has almost the same location as G20H1 shown in 22R sequence.

As per statement 'A16-A19-T18 base stacking organization, which is connected by an A17 linker' on p. 10 of the main text, numbering of selected residues in Fig. 4 may be helpful to assist a potential reader.

Response: The comment is accepted. We numbered the selected residues in Figs. 4 and 5.

Statement 'that berberine and coptisine can stabilize the physiological relevant KRAS-G4 in the human genome DNA context' needs to be rephrased as it represents interpretation of intensity of bands on a PAGE rather than actual KRAS-G4 formation in cells.

Response: The comment is accepted. We have amended the text in Page 16. "The results suggest that berberine and coptisine may be able to stabilize the physiological relevant KRAS-G4 and ...".

DMSO is used as a negative control in qRT-PCR and supposedly gives no inhibition. However, addition of DMSO could disrupt (reduce) KRAS-G4 formation. It is not clear from results in the current version of manuscript if authors have considered such an effect on destabilization of G4 and subsequent functional consequences. How significant is partial unfolding of KRAS-G4 in the presence of DMSO and how does it affect relative KRAS mRNA expression levels?

Response: Thanks for the comments. We tested the effect of DMSO on KRAS-G4 structure using NMR titration experiments. As seen in the new Supplementary Fig. 3, DMSO only has a negligible effect on KRAS-G4 formation. We conducted additional cellular activity testing and did not see a significant effect of DMSO on KRAS mRNA expression levels and the colony formation numbers (Supplementary Figs. 39 and 40).

Why was berberine (and coptisine) dissolved in DMSO when solubility in water can achieve over 5 mM concentration at room temperature? DNA concentration was 0.15 mM (Fig. 1c) which should make it possible to perform titration with NMR in completely aqueous solution.

Response: Thanks for the comments. We collected new NMR titration experiments with berberine and coptisine in completely aqueous solution, which showed almost identical spectra as previous collected with partial DMSO- d_6 (< 1.5%) (Fig 1c and Supplementary Fig. 2). We normally use DMSO- d_6 to prepare compound stock solution as DMSO- d_6 is hardly to evaporate, which keep the compound's concentration at constant for a longer time compared to water.

Minor.

- Footnotes of Fig. 2 need to be expanded to provide sufficient details.

Response: The comment is accepted. We provided sufficient details in Fig.2.

- Somewhat surprisingly 'various 2D NMR spectra at 3:1 berberine-DNA ratio were collected', but respective spectrum in Figure 6 was taken at 2:1 ratio. Is this a typo?

Response: Thanks for the comments. We corrected the footnotes in Fig. 6.

- Authors may wish to modify Fig. 8 to better illustrate introduction of additional side chains with increased resolution of suggested interactions with grooves.

Response: The comment is accepted. We modified Fig. 8 with increased resolution.

- Some grammatical corrections may be needed throughout the manuscript. As an example, the first sentence of Discussion section on p. 17 needs to be finished or connected with the next one, removing full stop in between.

Response: Thanks for the comments. Our manuscript has been edited with the assistance of native speakers and senior professors, and we believe the language now is much improved.

- Are G-tetrad planarity restraints really needed? An attempt should be made to release them, at least in the final stages of SA calculations. And report the results, whatever they show.

Response: Thanks for the comments. We re-calculated the structures without G-tetrad planarity restraints for molecular dynamics refinement. The obtained results were shown in Supplementary Fig. 42 and we did not see much difference compared to the G-tetrad planarity restrained structures.

Reviewer #3 (Remarks to the Author):

Wang et al report a structural analysis of the interaction of a G-quadruplex DNA formed in the promoter region of the KRAS oncogene with the small molecules berberine and coptisine. They use NMR, CD and EMSAs to characterize the binding and then use the KRAS1234 G4 with flanking regions for NMR structure determination using 1H NMR spectra only.

Functional effects of the two small molecules are shown by DNA polymerase assays in the absence and presence of inhibitors using a DNA template with the KRAS G4 motif and qRT-PCR monitored effects of transcription in lung cancer cells. The authors then discuss potential modification of the small molecules to improved binding and inhibition, however, not experimental verification is done.

The manuscript is interesting and studies a biomedically important topic. Identifying and optimizing specific inhibitors of DNA G4 quadruplexes is interesting, but also challenging. The presented structural work contributes to this. The work appears technical sound, although some more experimental details and additional experimental support should be provided. The activity in cells seems to be consistent with the proposed inhibition, but given the relatively weak binding affinity there is concern how specific the compounds indeed can function in cells.

Response: We appreciate the reviewer for these insightful comments. To address the

specificity of the compounds to the cells, we tested the cytotoxicity of the two compounds to the normal human lung epithelial cells (BEAS-2B). The berberine and coptisine do not show significant cellular toxicity to the BEAS-2B cells (Supplementary Figs. 37, 39, and 40).

To support the structural analysis the structure-based optimization of the small molecules is discussed only. However, this should be performed experimentally and validated by NMR and the functional assay to support the conclusions made and increase the impact of the work.

Response: This is a good suggestion for our future work. However, it is a complex task and goes beyond the scope of the current manuscript. We have shortened the relevant parts in the main text to avoid potential confusion.

NMR chemical shifts, restraints and coordinate files must be deposited in BMRB and PDB and accession codes provided in the manuscript.

Response: Thanks for the comments. We have deposited the experimental details at the Protein Data Bank and RMRB ((7X8N, 36476, free KRAS-G4), (7X8M, 36475, berberine-KRAS-G4), and (7X8O, 36477, coptisine-KRAS-G4)), and included the relevant information in Page 26.

Specific comments;

- Please mention K_d determined in the text (not only referring to Fig. 2c)

Response: Thanks for the suggestion. We included the K_d values in the main text. "The determined K_d values are 0.50 μM and 0.55 μM for berberine and coptisine, respectively (Fig. 2c)."

- The stoichiometry of the complexes seems not fully clear, saturation in the NMR titrations is mentioned at 3:1, but the CD seems to indicate 2:1? How do the authors rationalize the additional spectral change of imino signals from 2:1 to 3:1 (Fig. 1c), could this not indicate additional contacts and/or conformational changes beyond the 2:1 complex?

Likewise: the melting points increase significantly from 2 to 4 eq (Fig. 2c), how is this explained?

Response: Thanks for the comments. See also the response to referee 2, point 2. To be clear, the binding stoichiometry herein is 1:2. In Figure 1c, only minor changes were observed for the imino protons from 2 to 3 drug equivalents, where the G15-H1 is clearly seen at 3 drug equivalents. T_m changes were used to corroborate the binding stoichiometry of 1:2. For ligand binding at the intermediate-to-fast exchange on the NMR timescale, a ligand:DNA ratio higher than its binding stoichiometry is often needed to push the binding equilibrium toward the formation of a well-resolved complex, which shows 12 sharp imino proton peaks (*J. Med. Chem.* 2021, 64, 16205; *J. Am. Chem. Soc.* 2019, 141, 11059;

Angew. Chem. Int. Ed. 2016, 55, 12508). As large percentage of stable 2:1 binding complex structures were formed at higher ligand ratio, which is offered for further thermal stabilization. We added these details in Page 8.

- p.12: the authors speculate that competition between an A-T basepair and the A-berberine plane may explain that an unstable 3'-end complex is formed at 2:1 drug ratio. This is not clear, please explain and provide some experimental support.

Response: Thanks for the comments. After careful consideration, we think our speculation could be wrong because similar phenomena were seen for different systems in the 3'-end ligand-DNA complexes (*J. Med. Chem.* 2021, 64, 16205; *J. Am. Chem. Soc.* 2019, 141, 11059). We thus removed this sentence from the main text.

- Can the authors provide some experimental support, chemical shift changes, solvent exchange rate differences, J-couplings for the proposed charged and/or H-bond interactions?

Response: The charge and H-bond interactions are proposed based on the determined structure. We revised the statement and call them as "potential H-bond" and "potential electrostatic interactions" (Page 15).

We included berberine and coptisine proton chemical shift changes in Supplementary Figs. 29 and 30. We also collected the solvent exchange experiments. The data showed that guanine imino protons in ligand-KRAS-G4 complex structures are significantly protected compared to free KRAS-G4, which could be attributed to the π -stacking, electrostatic, and/or H-bond interactions between the ligand and KRAS-G4 (Supplementary Fig. 35). We added the information in Page 16.

- How were NOEs calibrated, how was spin diffusion considered?

Response: Chemical shift calibration was done indirectly for ^{13}C relative to DSS and directly for ^1H according to the water signal relative to DSS. We included the information in Page 20.

As all NOEs were cross-checked by different NOESY spectra at temperatures of 20, 25, and 35 °C with mixing times of 80, 150, 300, and 350 ms in 90% H₂O/10% D₂O solutions (Page 20), we do not think there are any spin diffusion problems.

We used a semi-quantitative method to obtain our distance restraints. Peaks are compared at different mixing times and categorized accordingly. Since they are not integrated and larger boundaries are used, calibration and correction for spin diffusion are not necessary. This method is widely used in the G4 field and also for other biomolecules.

- The ligand signals and intermolecular NOEs are severely line-broadened (Suppl Fig. 8,

11), - how could the authors analyze and integrate these NOEs for structural analysis.

Response: Thanks for the comments. You are right. Indeed, the ligand signals and intermolecular NOEs are severely line-broadened at 25 °C (new Supplementary Figs. 20 and 25). However, the ligand signals and intermolecular NOEs are much improved at 35 °C (Fig. 6 and Supplementary Figs. 18-19 and 23-24). We mainly used 35 °C NMR spectra for NOE-distance restraints. Moreover, we did not integrate these NOEs. The berberine-DNA and coptisine-DNA intermolecular cross-peaks were classed as very weak (6.0 ± 1.5 Å), weak (5.0 ± 1.5 Å), and medium (4.0 ± 1.5 Å) according to the spectra collected at different mixing times (Page 21).

- Please provide 2D NMR spectra superpositions showing the differences in chemical shifts of small molecules free and bound to the G4.

Response: The comment is accepted. We included the suggested 2D NOESY spectra in Supplementary Figs. 19 and 24.

- How were torsion-angle restraints derived from which experimental data?

Response: The torsion-angle restraints for the glycosidic angles were derived from the intensities of H1'-H6/H8 NOE cross-peaks and the corresponding C6/C8 chemical shifts of the HSQC experiments (Page 10). No other torsion-angle restraints were used.

- The DNA polymerase stalling occurs only above 40 uM berberine concentration, how is this consistent with the binding affinity of 0.5 uM?

Response: The DNA polymerase stop assay (Page 23-24) used a FAM-labeled DNA primer and the enzyme reaction was performed for 30 min (enzyme reaction time will affect the band intensity of the stalling product). The stalling products were subsequently visualized by scanning on a Bio-Rad ChemiDoc XRS+ System. Fluorescence measurements (Page 23) for K_d determination used a Jasco-FP8300 spectrofluorometer and directly measured the ligand-induced fluorescence intensity. These conditions are quite different between the two assays, especially for the detection sensitivity and the enzyme reaction time. Therefore, the discrepancy is reasonable when considering the different conditions in our opinions.

- Can the authors provide evidence that the G4 is indeed formed in the template for the DNA polymerase assay, and if possible also in the cell-based functional assay?

Response: We confirmed the G4 is indeed formed in the template for the DNA polymerase assay by the NMR experiment (Page 16, Supplementary Fig. 36). We are very sorry that we are unable to confirm it in the cell-based functional assay at current study.

- English language should please be checked, some phrases/wording sounds awkward (i.e

“hotly” line 9 of the introduction; “Lots of intramolecular NOEs”, p.11, ...)

Response: Thanks for the comments. Our manuscript has been edited with the assistance of native speakers and senior professors, and we believe the language now is much improved.

REVIEWER COMMENTS

Reviewer #1 (Remarks to the Author):

Remarks: The modifications seem to make the work and the article more robust and readable for non-experts.

Correction to be addressed:

For figure S15 and S18 the NOESY spectra in the top left corner seem to have folded or inserted the diagonal imino region. Please correct that, only experts know what is done. If you insert a small window with the imino region you need to put the ppm and refer to that modification in the legend.

Reviewer #2 (Remarks to the Author):

The authors responded to all of the reviewer's comments in a satisfactory manner. They even sought the advice of experts in the field for advice on the points raised and provided detailed responses.

Reviewer #3 (Remarks to the Author):

The authors have responded to my comments, changed/removed some statements, but added little new data. Some concerns remain regarding the specificity and molecular details of the ligand interactions and the lack of additional data to demonstrate target engagement in cellular assays. With half micromolar affinity and considering the molecular structures of the inhibitors it is expected that there will be significant off-target effects. Some SAR based on high-resolution structural analysis would be important to support the conclusions made and increase the novelty and impact of the work. Overall, the are interesting structural data of G4 inhibitor complexes reported and overall technical aspects are fine. Thus although the specificity of the compounds concerning their cellular activity remains unclear I would not object publication.

Specific comments:

- The authors state that NOE buildups have been measured and that there is not "worry" about spin diffusion, and that qualitative distance calibration is done in the "G4 field. These statements are worrying. The authors should show some representative NOE buildups for key NOEs and justify the distance calibration at the mixing time chosen. That something is done in the field does not mean it is correct
- I am not really convinced about the low activity in the polymerase assay compared to the stated higher affinity of the compounds.

Responses to reviewers' comments

Reviewer #1 (Remarks to the Author):

Remarks: The modifications seem to make the work and the article more robust and readable for non-experts.

Correction to be addressed:

For figure S15 and S18 the NOESY spectra in the top left corner seem to have folded or inserted the diagonal imino region. Please correct that, only experts know what is done. If you insert a small window with the imino region you need to put the ppm and refer to that modification in the legend.

Response: We appreciate the reviewer for these insightful comments. We have revised Figures S15 and S18 as suggested.

Reviewer #2 (Remarks to the Author):

The authors responded to all of the reviewer's comments in a satisfactory manner. They even sought the advice of experts in the field for advice on the points raised and provided detailed responses.

Response: We appreciate the reviewer for these insightful comments. Thanks.

Reviewer #3 (Remarks to the Author):

The authors have responded to my comments, changed/removed some statements, but added little new data. Some concerns remain regarding the specificity and molecular details of the ligand interactions and the lack of additional data to demonstrate target engagement in cellular assays. With half micromolar affinity and considering the molecular structures of the inhibitors it is expected that there will be significant off-target effects. Some SAR based on high-resolution structural analysis would be important to support the conclusions made and increase the novelty and impact of the work. Overall, the are interesting structural data of G4 inhibitor complexes reported and overall technical aspects are fine. Thus although the specificity of the compounds concerning their cellular activity remains unclear I would not object publication.

Response: We appreciate the reviewer for these insightful comments. We agree with the reviewer that the specificity of the compounds in cellular activity is weak and SAR analysis based on high-resolution structures would be important and necessary. We are working on that. Moreover, we added the following caveats in the discussion part: "Notably, target specificity of the two compounds appears to be weak and an in-depth structure-activity-relationship (SAR) analysis based on the determined high-resolution complex structures are highly required."

Specific comments:

- The authors state that NOE buildups have been measured and that there is not “worry” about spin diffusion, and that qualitative distance calibration is done in the “G4 field. These statements are worrying. The authors should show some representative NOE buildups for key NOEs and justify the distance calibration at the mixing time chosen. That something is done in the field does not mean it is correct.

Response: We thank the reviewer for these insightful comments. We included some representative NOE build-up curves in Supplementary Fig. 42. The NOE distance calibration was done using the cytosine H5/H6 distance (2.4 Å) (Page 20). Moreover, we added some details in the methods parts (Page 21): “For instance, in 350 ms spectrum, the intra-residue NOE cross-peaks of sugar H1′/H2′, H1′/H2′′, H1′/H4′, thymine H6/HMe, and cytosine H5/H6 are defined as strong NOEs; the intra-residue NOE cross-peaks of sugar H1′/H3′, H4′/H2′, H4′/H2′′, and anti-conformation H1′/H8 are defined as medium NOEs. The other NOE cross-peaks are classed based on intensities compared to these well-defined NOEs.” We included several references for the corresponding method parts.

- I am not really convinced about the low activity in the polymerase assay compared to the stated higher affinity of the compounds.

Response: Thanks for the comments. We believe the polymerase stop assay is much more complicated than the direct Kd measurement. The mechanism of how DNA polymerase overcomes the physical obstacle of G4 is not clear yet, we thus will keep this comment in mind in future studies.